# The Emerging Role of Polyphenols in the Management of Type 2 Diabetes

**DOI:** 10.3390/molecules26030703

**Published:** 2021-01-29

**Authors:** Yao Wang, Hana Alkhalidy, Dongmin Liu

**Affiliations:** 1Department of Human Nutrition, Foods and Exercise, College of Agricultural and Life Sciences, Virginia Tech, Blacksburg, VA 24060, USA; yaow@vt.edu; 2Department of Nutrition and Food Technology, Jordan University of Science and Technology, Irbid 22110, Jordan; haalkhalidy@just.edu.jo

**Keywords:** polyphenol, GLP-1, glucose homeostasis, microbiota, type 2 diabetes, obesity

## Abstract

Type 2 diabetes (T2D) is a fast-increasing health problem globally, and it results from insulin resistance and pancreatic β-cell dysfunction. The gastrointestinal (GI) tract is recognized as one of the major regulatory organs of glucose homeostasis that involves multiple gut hormones and microbiota. Notably, the incretin hormone glucagon-like peptide-1 (GLP-1) secreted from enteroendocrine L-cells plays a pivotal role in maintaining glucose homeostasis via eliciting pleiotropic effects, which are largely mediated via its receptor. Thus, targeting the GLP-1 signaling system is a highly attractive therapeutic strategy to treatment T2D. Polyphenols, the secondary metabolites from plants, have drawn considerable attention because of their numerous health benefits, including potential anti-diabetic effects. Although the major targets and locations for the polyphenolic compounds to exert the anti-diabetic action are still unclear, the first organ that is exposed to these compounds is the GI tract in which polyphenols could modulate enzymes and hormones. Indeed, emerging evidence has shown that polyphenols can stimulate GLP-1 secretion, indicating that these natural compounds might exert metabolic action at least partially mediated by GLP-1. This review provides an overview of nutritional regulation of GLP-1 secretion and summarizes recent studies on the roles of polyphenols in GLP-1 secretion and degradation as it relates to metabolic homeostasis. In addition, the effects of polyphenols on microbiota and microbial metabolites that could indirectly modulate GLP-1 secretion are also discussed.

## 1. Introduction

Type 2 diabetes (T2D) is becoming a global health problem [1], presently affecting 34.2 million (or 10.5%) people in the U.S. alone [2], with 90% of them being overweight or obese [3]. It is well-established that T2D results from insulin resistance in insulin sensitive tissues and subsequent pancreatic beta-cell dysfunction [4,5]. Therefore, most of the therapeutic interventions for T2D focus on lowering blood glucose levels by stimulating insulin secretion, decreasing hepatic glucose production, or enhancing insulin action at target tissues [6,7]. Gastrointestinal (GI) tract plays a primary role in regulating glucose homeostasis, in particular, postprandial blood glucose levels. In addition to digesting and absorbing nutrients, the intestine secretes hormones and generates neuronal signals in response to the ingested nutrients that coordinately exert profound metabolic effects [8,9,10,11]. Of these, the incretins, which are a group of metabolic hormones that have an insulinotropic action, have drawn considerable attention for developing novel therapeutic strategy for treating T2D [12]. Incretin hormones accounts for over 50% of postprandial insulin secretion in healthy individuals [13]. Notably, glucagon-like peptide-1 (GLP-1), an incretin secreted from enteroendocrine L-cells after food ingestion [14], plays a pivotal role in glucose homeostasis via potentiating glucose stimulated insulin secretion (GSIS) and promoting β-cell proliferation and survival [15,16,17]. In addition, GLP-1 delays gastric emptying, induces satiety, and reduces body weight in animal models of obesity and obese humans [18,19]. In addition, the gut microbiota, a large and diverse microbial community in the colon, is involved in metabolizing dietary components, primarily indigestible carbohydrates and phenolic compounds [20], thereby, affecting host’s biological activity at the level of the intestine and the whole body. Indeed, increasing evidence shows that the gut microbiota play an important regulatory role in metabolic homeostasis of the host, as alterations in the gut microbiota, termed dysbiosis, are associated with obesity and the pathogenesis of T2D [21].

A mounting body of evidence from preclinical studies indicates that dietary polyphenols, present in many fruits, vegetables, and medicinal herbs, may exert anti-diabetic effects [22,23,24,25]. A recent epidemiological study showed that intake of polyphenol-rich diet was associated with a lower risk of diabetes [26]. Likewise, clinical trials showed beneficial effects of polyphenols on glucose metabolism. For instance, daily intake of apple (*Malus pumila* cv. *Fuji*) polyphenols for 12 weeks significantly improved glucose tolerance in humans [27]. While various potential mechanisms for the anti-diabetic effects of polyphenolic compounds have been proposed, they are largely based on the in vitro studies with the use of pharmacological doses that often yield confounding or inconsistent results. For example, naringenin, a predominant flavanone from citrus fruit, was found to increase glucose uptake in cultured skeletal muscle cells [28], but suppress intestinal glucose absorption [29]. Similarly, the flavonol quercetin increased glucose uptake of muscle cells but inhibited the uptake in adipocytes and intestinal cells [30,31,32]. In these studies, the lowest effective concentration of these polyphenolic compounds for achieving the observed results was at least 10 µM. However, the circulating concentrations of most polyphenols, including their metabolites following dietary intake, are typically less than 5 µM [33], suggesting that many in vitro findings from using pharmacological doses of polyphenols might be physiologically irrelevant. Thus, the exact anti-diabetic mode and the major target tissues of polyphenols in vivo have not been fully elucidated.

Given the low bioavailability but relatively high concentrations of polyphenols in the GI tract following dietary ingestion, it is likely that the intestine might be a primary site for polyphenols to exert the observed metabolic actions [34]. In that regard, polyphenols may influence glucose homeostasis partially through modulating glucose and fatty acid digestion and absorption [35]. Interestingly, emerging evidence suggests that phenolic compounds also affect gut hormone release and microbiota composition, which are then directly or indirectly involved in regulating metabolic homeostasis. For example, it was shown that intake of coffee polyphenol increases postprandial GLP-1 concentration [36], which was associated with the lower risk of T2D in the individuals consuming 6 or more cups of coffee daily [37]. Epigallocatechin gallate (EGCG), a main polyphenol from green tea (*Camellia sinensis*), suppresses the secretion of orexigenic peptide hormone, ghrelin [38], thereby reducing food intake in rodents [39]. Polyphenol-enriched beverage that made from turmeric increases the postprandial concentration of anorexigenic hormone, peptide tyrosine tyrosine (PYY) in the healthy individuals [40], suggesting a potential anti-obesity action of turmeric polyphenols [41]. In this review, we first provide an overview about GLP-1 and the regulatory mechanisms for its secretion, and summarize recent developments in the effects of polyphenolic compounds on GLP-1, with emphasis on the signaling pathways activated by polyphenolic compounds in the regulation of GLP-1 secretion. In addition, the effect of polyphenols on microbiota and subsequent microbial metabolites is discussed as they relate to GLP-1 secretion. Lastly, emerging literature exploring the potential of polyphenols as dipeptidyl-peptidase-IV (DPP-IV) inhibitors is also highlighted.

## 2. Regulation of GLP-1 Secretion

### 2.1. GLP-1 Physiology

The gut is the largest endocrine organ in the body and releases an array of hormones, which play many key roles in physiological and metabolic regulation [42]. Specifically, GI hormones secreted after a meal plays a critical role in regulating appetite, food intake, and glucose homeostasis [11]. GLP-1 is a well-studied incretin hormone secreted by L-cells in the gut [43], the second largest population of enteroendocrine cells [16,44]. GLP-1 is produced from post-translational processing of the proglucagon gene in the L-cells [45], leading to the generation of two major forms of GLP-1, GLP-1-(7–37) and GLP-1-(7–36)NH_2_, which are then released into the bloodstream [46]. GLP-1 has a very short half-time (2–3 min) in the blood because it is rapidly degraded by DPP-IV that cleaves GLP-1 from its N-terminal into GLP-1-(9-37) and GLP-1-(9-36) NH_2_ [43]. 

The L-cells are distributed throughout the intestine with much higher density in the ileum and the colon. The polarized morphology of L-cells allows its apical surface to face the lumen of the intestine thus facilitating sensing of nutritional, neural, and hormonal signaling by the cells [18,47]. The role of GLP-1 in maintaining glucose homeostasis is mediated through its receptor (GLP-1R). GLP-1 binding to its receptor triggers cyclic adenosine monophosphate (cAMP) signaling that augments GSIS from pancreatic β-cells while inhibiting glucagon release from α-cells [48]. In addition, the activation of GLP-1 signaling promotes β-cell proliferation and survival, therefore preserving islet mass [49]. Furthermore, GLP-1 slows the gastric emptying and gut mobility, and probably also targets its receptor in the hypothalamus to promote satiety, thereby reducing food intake [50,51]. Impaired GLP-1 secretion in response to intake of meals or glucose has been reported in obese and T2D individuals [52,53,54,55]. However, a systematic analysis of GLP-1 secretory data from human studies concluded that patients with T2D, particularly those with relatively low HbA_1c_ did not show reduced GLP-1 secretion in response to glucose or a mixed meal [56]. In addition to ingested nutrients, GLP-1 secretion could also be influenced by other factors such as sex [55], obesity, insulin resistance, β-cell function, and diabetes state [57]. Therefore, it remains to be determined whether the defect of nutrient-induced GLP-1 secretion occurs early and is responsible for the pathogenesis of T2D. Regardless, activation of the GLP-1 signaling system is effective for the treatment of T2D, while also exerting other beneficial effects, such as promoting weight loss and β-cell function, and improving cardiovascular parameters [58]. Several GLP-1R agonists have been approved by the FDA for treating T2D. The first GLP-1R agonist, exenatide, also known as Byetta, shared 55% homology to human GLP-1, and when administered with a dose of 5 µg, it significantly improved hyperglycemia in T2D patients [59]. Consistently, another recently developed GLP-1R agonist, liraglutide, which has 96% homology with human GLP-1, showed more efficacious glycemic control as compared with that of exenatide [60]. 

### 2.2. Neuronal and Hormonal Regulation of GLP-1 Secretion

GLP-1 secretion is primarily stimulated by ingested nutrients. It has been observed that GLP-1 is secreted into the circulation in a biphasic pattern, with an initial rapid rise in circulating GLP-1 levels occurring 15 to 30 min after food ingestion, followed by a prolonged second phase about 60 min later [61]. Neuroendocrine regulators rather than the direct sensing of L-cells to the nutrients, should be primarily responsible for the early rise of GLP-1 levels because the digested nutrients are unable to reach the ileum or colon within 30 min where L-cells are primarily located [62]. Indeed, it was found that nutrients in the proximal intestine are able to trigger GLP-1 secretion from the distal gut via activation of vagus nerves [63,64]. In a study performed in rats, it was found that left cervical vagotomy reduced the basal GLP-1 concentration [63], suggesting that vagus nerve plays a role in maintaining basal GLP-1 release. In addition, infusion of nutrients into the proximal duodenum failed to increase GLP-1 secretion in rats with subdiaphragmatic vagotomy [63], confirming that vagus nerve is responsible for the first phase of postprandial GLP-1 secretion. Acetylcholine, a well-known neurotransmitter, is involved in the vagal nerve activation-induced GLP-1 secretion [65]. In that regard, acetylcholine released from vagal efferent nerves stimulates GLP-1 secretion via its receptors, muscarinic receptors, which are expressed on the L-cells [66]. There are reportedly two types of muscarinic receptors (M1 and M2) on the surface of L-cells. M1 muscarinic receptor is coupled to Gα_q/11_ and its activation triggers phospholipase C (PLC)/inositol trisphosphate (IP_3_)/Ca^2+^ signaling leading to GLP-1 release, while the M2 muscarinic receptor is coupled to Gα_i_, and its activation inhibits adenylyl cyclase and attenuates the cAMP signaling pathway [67], suggesting that M1 receptor plays a major role in mediating acetylcholine effect on GLP-1 secretion, as activation of M2 muscarinic receptor could potentially inhibit GLP-1 secretion via suppressing cAMP signaling [67]. 

Other gut hormones may affect GLP-1 secretion. Glucose-dependent insulinotropic peptide (GIP), which is secreted from the small intestinal K-cells in response to nutrient intake [43], may also be involved in the regulation of GLP-1 secretion. GIP-induced GLP-1 secretion might, in part, depend on the activation of the vagus nerve. In rodents, the administration of GIP at a supraphysiological dose increased GLP-1 release, however, this effect was abrogated by hepatic branch vagotomy [63]. In addition, the GIP may directly stimulate GLP-1 release via its receptor on L-cells, as demonstrated in cultured fetal rat intestinal cells and GluTag L-cells [68]. In addition to GIP, ghrelin, an orexigenic gut hormone mainly released from the stomach, was reported to enhance postprandial GLP-1 secretion. The plasma concentration of ghrelin increases during fasting and decreases after refeeding [69]. In clonal L-cells, ghrelin triggered GLP-1 secretion via activation of the ghrelin receptor and extracellular signal-related kinase 1/2 (ERK1/2)-dependent pathway [70]. In vivo, the intraperitoneal administration of ghrelin enhanced glucose-stimulated GLP-1 secretion (GSGS) and improved glucose tolerance in mice, while pharmacological inhibition of ghrelin receptor blocked ghrelin-induced GLP-1 secretion [71]. Consistently, ghrelin-induced improvement in glucose tolerance was diminished in mice administered with a GLP-1R antagonist and in GLP-1R-deficient mice [71]. These outcomes suggest that the effect of ghrelin on glucose tolerance is mediated through the GLP-1/GLP-1R pathway. Such an interaction between ghrelin and GLP-1 secretion in response to fasting and refeeding demonstrates a precise and reciprocal regulation of food intake in the body. In that regard, the increase in ghrelin levels during fasting stimulates appetite and thus increases food intake. Subsequently, the ingestion of food stimulates GLP-1 secretion, which in turn reduces appetite and decreases ghrelin secretion [72]. 

### 2.3. Nutritional Regulation of GLP-1 Secretion

As discussed above, GLP-1 is secreted in a biphasic pattern in response to the ingested meal. The second phase of GLP-1 secretion is largely stimulated by the digested macronutrients via their respective receptors on the plasma membrane of L-cells [73,74] (Figure 1). Sweet taste receptors 1 (Tas1Rs) have been reported to mediate monosaccharide-induced GLP-1 secretion [75]. Tas1Rs are present in the GI tract of both rodents and humans [75,76]. The activation of these receptors by glucose on the L-cells triggers GLP-1 secretion via the PLC/IP_3_/Ca^2+^-sensitive transient receptor potential channel M5 (TRPM5) pathway [77]. Consistently, blockage of Tas1R2/Tas1R3 reduced the increase in circulating GLP-1 concentrations in response to glucose administration in humans [78]. In addition, glucose could induce GLP-1 secretion in a Tas1R-independent pathway, where the glucose is transported into the cells via sodium-dependent glucose transporter 1 (SGLT1) and undergoes glycolysis and oxidation, resulting in increased ATP production and subsequent closure of K_ATP_ channels, which sequentially causes depolarization of the plasma membrane, opening of voltage-gated L-type Ca^2+^ channels, Ca^2+^ influx, and ultimate GLP-1 secretion [79,80]. Thus, inhibition of glucose uptake could prevent the cascade of these events and ablate GSGS [81].

Fatty acids (FAs)-stimulation of GLP-1 secretion is largely mediated via their respective receptors, which belong to G-protein coupled receptors (GPCRs). Specifically, four FA receptors (FFARs), FFAR1 (GPR40), FFAR2 (GPR43), FFAR3 (GPR41), and FFAR4 (GPR120), which are coupled to G-proteins, are involved in FA-induced GLP-1 secretion [73,74]. The binding of FFAR1 or FFAR4 by their ligands, mainly medium to long chain FAs, like linoleic acid [82,83] and omega-3 polyunsaturated FAs (PUFAs) [84], activates Gα_q_, which stimulates PLC to cleave membrane-associated phosphatidylinositol 4,5-bisphosphate (PIP2) into IP_3_ and diacylglycerol (DAG). The released IP_3_ binds to the IP_3_ receptor on the ER, triggering Ca^2+^ release from the ER and subsequent GLP-1 secretion [85]. In addition, DAG can also induce GLP-1 secretion through activation of the protein kinase C (PKC)/protein kinase D (PKD) pathway [86]. FFAR2 and FFAR3 are activated by short-chain FAs (SCFAs), in particular, acetate, propionate, and butyrate [74,87]. These SCFAs are generated from microbial fermentation of non-digestible dietary nutrients, such as fiber and other indigestible carbohydrates [88]. FFAR2 is coupled to both Gα_i_ and Gα_q_, whereas FFAR3 is only coupled to Gα_i_ and activation of FFAR3 in CHO cells overexpressing FFAR3 resulted in Gα_i_-mediated suppression of intracellular production of cAMP [89]. It was shown that mice deficient of FFAR2 had a blunted GLP-1 secretion in vivo [74]. Similarly, FFAR3-deficient mice also displayed impaired GLP-1 secretion in response to the administration of SCFAs [74], suggesting that both FFAR2 and FFAR3 play a role in SCFA-induced GLP-1 secretion. Although the mechanism is still unclear, it is possible that the impaired GLP-1 secretion in FFAR3-deficient mice could be due to the reduced FFAR2 expression [74]. However, more recent work using the perfused rat colon demonstrated that specific activation or blockage of FFAR2 or FFAR3 had no effect on colonic GLP-1 release, suggesting that SCFAs may trigger GLP-1 secretion independent of the FFAR2 and FFAR3 [90]. Therefore, it remains to be determined whether L-cell-specific FFAR2/FFAR3 are involved in mediating SCFA-stimulated GLP-1 secretion.

The ingestion of high-protein diets also was shown to increase GLP-1 secretion in rodents and humans [91,92]. Protein-induced GLP-1 secretion is likely executed via its enzymatic products, peptides and amino acids (AAs), which might act through peptide transporter-1 (PEPT1) and calcium-sensing receptor (CaSR) to induce GLP-1 secretion [93]. PEPT1 has a rather broad substrate specificity, as it has been demonstrated to transport di and tripeptides that have L-AA, an acidic, or hydrophobic group at C-terminus, a weak basic group in α-position at N-terminus, a ketomethylene or acid amide bond, or a trans conformation of peptide bonds [94,95]. In addition to PEPT1, CaSR is involved in the peptide-induced GLP-1 secretion. CaSR is a G-protein-coupled receptor widely expressed in human tissues. In the intestine, CaSR is involved in nutrient sensing [96] and serves as a broad-spectrum of L-aromatic AA sensor [97]. Studies showed that L-phenylalanine and L-tryptophan induce GLP-1 secretion through the CaSR-mediated mechanism [98,99]. The activation of CaSR triggers depolarization of the cell membrane that leads to the opening of the voltage-gated Ca^2+^ channel and subsequent influx of Ca^2+^ [100]. However, not all AAs elicit Ca^2+^-mediated GLP-1 secretion via a CaSR-independent mechanism. L-glutamine was shown to induce [Ca^2+^]_i_ increase and GLP-1 secretion via a cAMP signaling-mediated but PLC-independent pathway [80], indicating that this AA-induced GLP-1 secretion is not associated with the activation of CaSR [101]. Interestingly, it was reported that L-cells also uptake L-glutamine through one of the Na^+^-dependent AA transporters, SLC38A2 [102]. Intracellular glutamine then enhances glutamate dehydrogenase and ATP production via promoting the citric acid cycle [103], which results in the depolarization of the cell membrane, triggering GLP-1 secretion. L-ornithine, a non-proteinogenic AA, can trigger GLP-1 secretion via activating the GPRC6A/Gα_q_/PLC/Ca^2+^ pathway [101].

## 3. Polyphenols

Polyphenolic compounds are a group of secondary metabolites that are ubiquitously present in many plants and foods. The biosynthesis of most polyphenols in the plant originates from the phenylalanine through a shikimate/phenylpropanoid pathway. However, secoiridoid polyphenols, typically derived from Oleaceae family, are synthesized through a mevalonic acid pathway [104]. Based on their diverse chemical structures, polyphenols are classified into phenolic acids, flavonoids, stilbenes, and lignans [105], their represent chemical structures are shown in Figure 2 The major food or plant sources of various polyphenols are beyond the scope of this review, but the interested readers can refer to previous review articles on this topic [106,107]. Phenolic acids are a group of compounds with a phenolic ring and carboxylic acid. Hydroxycinnamic acids are the most common phenolic acids in fruits [108], which primarily consist of caffeic acid, chlorogenic acid, and ferulic acid. Flavonoids are the largest and widely distributed group of polyphenols. The main structure of flavonoids is diphenylpropanes, consisting of two aromatic rings (A and B) linked by a heterocyclic ring C [109]. Based on the differences in the structures of the heterocyclic ring, flavonoids are further divided into six groups: flavones, flavonols, flavanones, isoflavones, flavanols, and anthocyanins. Stilbenes are a relatively small group of plant metabolites synthesized through the phenylpropanoid pathway, and they contain a 1,2-diphenylethylene as the basic structure. Stilbenes are primarily found in grapes and some berries, and one of the well-known stilbenes in human diet is resveratrol, which is present in red wine, grape skin, and blueberry. Lignans are derived from phenylalanine via the dimerization of substituted cinnamic alcohols. Flaxseed and sesame seed are very rich sources of lignans with matairesinols and secoisolariresinols being the dominant forms [110,111].

Most dietary polyphenols exist naturally in the form of esters, glycosides, or polymers, many of which are poorly absorbed by the intestine. One of the most extensively studied categories of polyphenols, are flavonoids. A majority of flavonoids are non-covalently bound to glucose in plants, and some are present as flavonoid oligomers such as epicatechin in cocoa. The hydrolysis of the glycoside form of flavonoids begins in the mouth where flavonoids are partially released from the food matrix by saliva [112]. The low pH in the stomach facilitates the degradation of flavonoid oligomers into small units [113], but most flavonoids are hydrolyzed and further metabolized in the small intestine [114,115], where β-glucosidases cleave the glucose moiety from flavonoids and generate the aglycone form [116]. The hydrophobic flavonoids pass the apical membrane via passive diffusion, while other flavonoids in the glycosylated form can be transported into the enterocytes via SGLT1 [117]. Afterward, the absorbed flavonoids are conjugated with glucuronic acid by uridine-diphosphate-glucuronosyltransferase in the enterocytes. The degree of conjugation depends on the structure of the absorbed flavonoids. For instance, flavonoids containing a hydroxyl group on the B-ring are less predisposed to glucuronidation, but those containing catechol are more prone to glucuronidation [118]. Flavonoids that are absorbed from the small intestine are transported to the liver where they can be further conjugated with sulfate and/or methyl groups and then partially excreted through bile [119,120,121]. A large proportion of ingested flavonoids do not undergo hydrolysis or conjugation in the small intestine [122] and are not absorbed [121]. Instead, these unabsorbed flavonoids reach the colon where they are metabolized by colonic bacteria into smaller molecules such as phenolic acids, which can be partially absorbed into the circulation [123,124]. Specifically, some intestinal bacteria strains have the enzymatic activity of glucuronidase and glucosidase that can deconjugate the glucuronidated or glycosylated forms of flavonoids, and yield simple deconjugated products that are partially absorbed and metabolized by colonic enterocytes.

### 3.1. Polyphenols and GLP-1 Secretion

Emerging evidence shows that some polyphenols can induce GLP-1 secretion from L-cells via various mechanisms Specifically, polyphenols can function as ligands of GPCRs, or regulate intracellular signaling molecules to modulate GLP-1 secretion. Additionally, some polyphenols might indirectly cause GLP-1 secretion mediated by gut microbiota. The review below summarizes the studies in the area of polyphenols and modulation of GLP-1 as well as the potential underlying mechanisms for this effect.

#### 3.1.1. Polyphenols as Activators of GPCRs

L-cells apparently could respond to a variety of structurally unrelated polyphenols via various mechanisms. Several polyphenolic compounds have been found to regulate GLP-1 secretion via GPCRs or cAMP signaling pathway, as summarized in Table 1. Specifically, GPCRs on the L-cells might be activated by polyphenols to regulate GLP-1 secretion. Curcumin, a polyphenolic turmeric, has been found to induce GLP-1 secretion in cultured L-cells with EC50 of 4.5 μM [125,126]. Oral intake of 1.5 mg/kg curcumin increased GSGS concomitant with improved glucose tolerance in rats [127]. It was further found that curcumin-induced GLP-1 secretion can be abrogated by the Ca^2+^ channel antagonist or IP_3_-receptor inhibitor, suggesting that both Ca^2+^ entry and release from intracellular store play a role in mediating curcumin-stimulated GLP-1 secretion, the latter of which may be due to the activation of PLC [127]. Interestingly, preincubation of GluTag L-cells with an antagonist (GW1100) of GPR40/120, both of which are coupled to Gα_q_ to activate the IP_3_/Ca^2+^ signaling pathway, blunted curcumin-induced GLP-1 secretion [127], demonstrating that curcumin-induced GLP-1 secretion could be mediated via the activation of GPR40/120. Consistently, administration of the antagonist of GPR40/120 abolished curcumin-induced GLP-1 secretion in rats [127]. While it is still unclear which receptor plays the predominant role in curcumin on GLP-1 secretion, the reported results suggest that curcumin could be a ligand of GPR40/120. Delphinidin 3-rutinoside (D3R), one of the anthocyanin derivatives from berry, increased GLP-1 secretion from L-cells with the lowest effective concentration of 10 μM, via a GPR40/120-mediated mechanism [128]. Similarly, the oral intake of 5 mg/kg of anthocyanin-rich blackcurrant (*Ribes nigrum L*.) extract, which contains 19.3% D3R, 4.6% delphinnidin 3-glucoside, 19.4% cyanidin 3-rutinoside, and 2% cyanidin 3-glucoside, augmented GSGS and improved glucose tolerance in rats [129]. Further, diabetic mice fed a diet containing 11 g/kg anthocyanins-rich extract from blackcurrant (*R. nigrum*) for 7 weeks showed an improved plasma GLP-1 concentration, accompanied with reduced hyperglycemia [130]. In contrast to the findings in animal studies, intake of anthocyanins-rich blackcurrant juice, which contains 599 mg anthocyanins (260 mg D3R, 209 mg cyanidin-3-rutinoside, 76 mg delphinidin-3-glucoside and 33 mg Cyanidin-3-glucoside), failed to elevate postprandial plasma GLP-1 levels in healthy individuals but led to a slightly decrease on GLP-1 level at 90 min after intake of anthocyanins-rich blackcurrant juice [131]. While the reasons for these inconsistent results observed in rodents and human studies are unclear, the dose of anthocyanins that was used in rats and in diabetic mice is equivalent to 57 mg/day and 12 mg/day, respectively, which are much lower than the dose used in above human study, implying that the supraphysiological dose might alter the stimulatory effect of anthocyanins on GLP-1 secretion. Other than flavonoids, a natural phenolic acid, grifolic acid, activated GPR120 that led to GLP-1 secretion from cultured L-cell at the concentration of 30 μM [132]. In addition, the elevated [Ca^2+^]_i_ was detected in grifolic acid-treated cells, suggesting that this compound induced GLP-1 secretion was mediated via GPR120/Ca^2+^ signaling. However, it is presently unknown whether grifolic acid stimulates GLP-1 secretion in vivo.

As GPR120/GPR40 are FA receptors, it is unclear how above discussed polyphenols are able to activate GPR120/GPR40. Interestingly, Cho et al. tested the effect of five different polyphenols (chrysin, pinocembrin, galangin, pinobanksin, and caffeic acid phenethyl ester) on GPR120 and found that flavanones (pinocembrin and pinobanksin) are more potent in activating GPR120 than that of tested flavones (chrysin and galangin) [133]. The only structure difference between these two subcategories of flavonoids is that flavones have a double bond between the C2 and C3 positions, whereas flavanones is saturated in C-ring, suggesting that the double bond between C2 and C3 might decrease the affinity of polyphenols for GPR120 136]. Indeed, curcumin and D3R, which activate GPR120 as aforementioned, also lack this double bond.

In addition to FA receptors, it was found that the superfamily of bitter taste receptors (Tas2Rs), which consist of 25 members in humans [134] and 35 members in rats and mice [135], are expressed in the intestinal enteroendocrine cells [136,137], as well as clonal L-cell lines [138]. Tas2Rs are GPCRs that detect a diversity of bitter compounds and transduce the signal through G-protein-coupled signaling. Like other GPCRs, activation of Tas2Rs increases GLP-1 secretion via the Gα-gustducin/Gβγ-triggered pathway, which can initiate IP_3_/Ca^2+^ signaling, subsequently leading to the opening of transient receptor potential (TRP) ion channels [139], and triggering Ca^2+^ influx and exocytosis of GLP-1 vesicle. Given that many dietary polyphenolic compounds are known to have a bitter taste, and they can interact with various taste receptors, in particular, Tas2Rs [140], it is possible that Tas2Rs mediate polyphenols-induced GLP-1 secretion. Tas2Rs can be activated by a wide variety of polyphenols, but it appears that different polyphenolic compounds interact with different or the same forms of TasRs depending on their chemical structures. For instance, Tas2R14, one of the human TasRs with a broad agonist spectrum [141], is activated by ester form of phenolic acids, such as ferulic acid ethyl ester [142], whereas anthocyanins bind to the human taste receptor subtypes Tas2R5 and Tas2R7, isoflavones and epigallocatechins have better affinity to Tas2R39 [143]. While polyphenols have been found to activate several Tas2Rs, studies on whether polyphenols stimulate GLP-1 secretion via Tas2Rs are limited. EGCG is one of the most effective Tas2R39 activator among polyphenolic compounds with an EC_50_ value of 8.5μM [142]. In addition, EGCG activated transient receptor potential ion channel A1 (TRPA1) in the mouse clonal L-cell line (STC-1) [144], which was opened after Tas2Rs activation to initiate the Ca^2+^ influx. EGCG has been reported to induce GLP-1 secretion in the mouse ileum segment [145], and oral intake of EGCG stimulated GLP-1 secretion in mice and diabetic patients [146,147], suggesting that EGCG-stimulated GLP-1 might be mediated via Tas2Rs. However, as EGCG activates multiple subtypes of Tas2Rs, such as Tas2R43 [142], which expresses on L-cells in both mice and humans [148], more studies are needed to determine which Tas2Rs are involved in the EGCG-induced GLP-1 secretion. Grape seed-derived proanthocyanidins (GSPE), which is composed of monomers or flavan-3-ols (21.3%), dimers (17.4%), trimers (16.3%), tetramers (13.3%) and oligomers (31.7%) of procyanidins, were found to increase GLP-1 secretion from clonal L-cells as well as mouse ileum and colon segments [149]. Consistent with this ex vivo finding, a single oral administration of 1 g/kg GSPE induced GLP-1 secretion in rats [149]. The stimulatory action of GSPE on GLP-1 secretion might result from a rise in intracellular Na^+^ influx via cation channel, TRPM5, which leads to the depolarization and the opening of voltage-gated calcium channel, resulting in an increase in Ca^2+^ influx [150], an essential step for exocytosis of GLP-1 granules. It is noted that TRPM5 is the common downstream molecule on the taste receptor signaling pathway, suggesting that the GSPE-induced GLP-1 release might depend on the taste receptor. As condensed tannins are ligands of Tas2Rs [142], proanthocyanidins might also activate taste receptors to regulate GLP-1 secretion. However, the degree of polymerization of proanthocyanidins might affect their affinity for taste receptors [143], for example, procyanidin trimer C2 activates Tas2R5, but procyanidin dimer B2 fails to activate any Tas2Rs. Thus, studies on specific proanthocyanidin from GSPE should be performed to verify the activity of certain proanthocyanidin on Tas2Rs.

GPCRs play a critical role in neuronal, hormonal, and nutritional regulation of GLP-1 secretion [151]. While polyphenols have been reported to activate multiple types of GPCRs [152], the most abundant GPCRs that expressed on enteroendocrine L-cells are FA receptors and taste receptors [153]. Therefore, efforts identifying the GPCRs that mediate polyphenols-induced GLP-1 secretion are primarily focused on FA receptors and taste receptors. As one polyphenol might be able to activate multiple receptors, in particular Tas2Rs, and most Tas2Rs share the same intracellular signaling to regulate GLP-1 secretion, the results from pharmaceutical inhibition of specific Tas2R could be misleading and thus should be carefully interpreted. In addition, most of these findings were obtained using cultured cells and tissues, and their physiological relevance as well as the most effective doses needed to be confirmed in vivo, given that ingested polyphenols can be metabolized and their actions can be affected by interacting with proteins in the GI tract.

#### 3.1.2. Polyphenols as Regulators of cAMP Signaling

It is well established that the elevation of intracellular cAMP levels induces both transcription and secretion of GLP-1 via protein kinase A (PKA) and Epac, a cAMP-activated effector protein [154,155]. Some polyphenols may promote GLP-1 secretion via activating cAMP-mediated intracellular signaling. Genistein, an isoflavone present in legumes and Chinese herbal medicines *Genista tinctoria Linn* and *Sophora subprostrala Chun* (Fabaceae family), is a popularly used supplement for various presumably beneficial effects. Indeed, it was shown that dietary intake of genistein or soy products lowered plasma glucose in diabetic animals [156,157] and in humans [158,159]. At physiologically achievable concentrations, genistein can activate cAMP signaling to induce insulin secretion from pancreatic β-cells [160]. In diabetic mouse models, dietary provision of genistein increased plasma insulin levels, and mitigated diabetes [161,162,163,164]. Interestingly, a recent study indicated that genistein enhanced GLP-1 secretion in alloxan-induced insulin deficient diabetic rats [165]. While this is a notable finding with respect of genistein effect, it is actually uncertain whether increased circulating GLP-1 levels as observed in genistein-treated diabetic animals were due to directly enhanced GLP-1 secretion, and whether cAMP signaling is involved in this genistein action.

Recently, it was found that hispidulin, a flavone found in Petasites, Artemisia, and Salvia, dose-dependently stimulates GLP-1 secretion in GluTag L-cells and primary mouse ileum crypts, with as low as 1 µM eliciting a significant effect [166]. In addition, oral administration of hispidulin significantly exerted antidiabetic effect concomitant with improved non-fasting insulin concentrations and β-cell survival in diabetic mice, consistent with well-established metabolic effects of GLP-1. It was further shown that hispidulin-induced GLP-1 secretion was mediated via increased intracellular cAMP content, likely as a result of the inhibition of phosphodiesterase (PDE), the enzyme that degrades cAMP into AMP [166]. However, it is unclear how hispidulin inhibits PDE in L-cells. Interestingly, this study further found that apigenin and luteolin, two other flavones, also increased GLP-1 secretion, whereas no other subclasses of tested flavonoids were able to significantly induce GLP-1 secretion from L-cells, suggesting that flavones, but not other subtypes of flavonoids, may be GLP-1 agonists. While the specific chemical structure on hispidulin or other flavones responsible for its stimulatory action on GLP-1 secretion is presently unclear, flavones have a double bond between C2 and C3 of the C-ring in the flavonoid skeleton, and an oxidized group at the C4 position of its B-ring, which may be crucial for this unique effect of flavone on GLP-1 secretion. Nevertheless, this research may have revealed a new mechanism underlying the antidiabetic effects of some flavones as observed in animal models [167,168].

Silymarin, a flavonolignan isolated from milk thistle (Silybum marianum), can stimulate GLP-1 secretion in obese rats [169]. While the mechanism underlying silymarin-induced GLP-1 secretion is unknown, it was found that this compound stimulates insulin secretion by inhibiting PDE activity in insulin secreting cell line [170], indicating that silymarin might have the similar effect in L-cells.

Coffee-derived polyphenols were also shown to stimulate GLP-1 secretion from human clonal L-cell in a dose-dependent manner, which was mediated via elevated intracellular cAMP signaling [36]. In line with this in vitro result, oral administration of coffee polyphenols to mice induced moderate but significant increase in GSGS [36]. Notably, a single intake of coffee-extract polyphenols, which mainly contain caffeoylquinic acids and chlorogenic acids, increased postprandial GLP-1 levels in healthy male adults [171]. Consistently, a large population cohort study showed that high coffee consumption was associated with low risks of T2D [172]. Caffeoylquinic acid and caffeic acid, two main polyphenols from coffee extract, have been found to inhibit PDE activity with IC_50_ of 0.49 and 0.48 mM, respectively [173], which might ascribe to the increased GLP-1 secretion following administration of coffee polyphenols.

### 3.2. The Potential Role of Microbiota in Polyphenol-Stimulated GLP-1 Secretion

In addition to directly targeting L-cells to stimulate GLP-1 secretion, polyphenols might also modulate GLP-1 secretion indirectly involving intestinal microbiota. The human gut hosts a vast diversity of microorganisms that have been the subject of extensive research in recent years, as they may play important roles in human physiology, metabolism, and disease. Before weaning, the composition of intestinal microbiota is simple, relatively unstable, and undergoes drastic changes [174], whereas the composition of gut microbiota in healthy adults is substantially stable over time [175,176]. However, dietary composition can profoundly influence the gut microbiota composition in the host [177,178,179]. As to nutrient metabolism, gut microbiota metabolizes non-digestible nutrients into SCFAs and gas. It was found that dietary intake of resistant starch, which escapes digestion in the small intestine, increased the proportion of butyrate-producing bacterium *Eubacterium hallii* [177], and sequentially increased butyrate production [180], suggesting that the shift in dietary pattern can modulate the abundance of SCFA-producing bacteria.

A large portion of the ingested polyphenols enters the large intestine where non-absorbed polyphenols are fermented by various colonic microbiota to generate a wide range of metabolites because of the large variability of their chemical structures [181]. For example, the flavonol, myricetin and quercetin, are structurally similar, and the only difference between them is that myricetin has three hydroxyl groups on the C ring while there are two for quercetin. Metabolism of quercetin by microbiota produces 2-(3,4-dihydroxyphenyl)-acetic acid and protocatechuic acid, but breakdown of myricetin generates 2-(3,5-dihydroxyphenyl)-acetic acid and gallic acid [182,183]. Resveratrol, a stilbenoid with two benzene rings, can be fermented by *Slackia equolifaciens* and *Adlercreutzia equolifaciens* to produce lunularin [184]. On the other hand, polyphenols can modulate the relative abundance of the gut microbiota. Some polyphenols have antibiotic activity by suppressing the growth of pathological bacteria. For instance, when intestinal bacteria were cultured with polyphenolic compounds from tea extracts, the growth of pathological bacteria, *Clostridium perfringens*, *Clostridium difficile*, and *Bacteroides spp* was inhibited, but the abundance of the probiotic bacteria was not affected [185]. In addition, some polyphenols can promote the growth of beneficial bacteria while inhibiting the growth of pathogens, therefore exhibiting a prebiotic-like effect. The polyphenols extracted from concord grapes and fermented green tea were shown to reduce high-fat-diet (HFD)-induced increase in the ratio of Firmicutes/Bacteroidetes and protect the intestinal barrier function in rodents [186,187]. Grape polyphenols, which primarily include anthocyanins, flavanols, flavonols, stilbenes and phenolic acids [188], enhanced the growth of *Akkermansia (A) muciniphila* in HFD-fed mice, which is an important species for degrading mucin [186].

As polyphenols modulate microbial abundance, they might also alter the production of microbial metabolites. As aforementioned, SCFAs are the main microbiota metabolites of non-digestible food components, and SCFAs, primarily acetate, propionate, and butyrate, can trigger GLP-1 secretion in vitro and in vivo via their GPCRs [74,87]. Clostridial clusters IV and XIVa, which belong to the Firmicutes phylum, are the two most important butyrate producers [189]. Among these two taxa, *Faecalibacterium* (F) *prausnitzii* and *Eubacterium (E) rectale* are the two most dominant bacterial strains in the human colon, and they play a major role in producing butyrate [190]. *F. prausnitzii* possesses butyryl coenzyme A (CoA): acetate CoA transferase and acetate kinase activities, and it converts two molecules of acetyl-CoA to butyrate [191], whereas *E. rectale* catalyzes butyrate formation from lactate and acetate [192]. In addition to butyrate-producing bacteria, there are bacteria species that produce other SCFAs, such as bifidobacterium species (Phylum Actinobacteria), which ferment carbohydrates to generate acetate [193], and *A. muciniphila* (Phylum Verrucomicrobia) that produces acetate and propionate [194,195]. Dietary polyphenols have been shown to enhance the population of SCFA-producing bacteria in fecal microbiota from humans [196] (Table 2). For instance, exposure of human fecal bacteria to 150 mg/L catechins enhanced the growth of *E. rectale*, the butyrate-producing bacteria, but this effect was not observed until after 17 h of incubation period [196]. As the regulation of bacterial growth by polyphenols might be mediated by their metabolites, it is possible that a longer incubation time may be needed for this strain of bacteria to generate adequate amounts of metabolites necessary for stimulating bacteria growth. Similar to this finding, in another study, the incubation of human fecal bacteria with 10 μg/mL chlorogenic acid for 48 h increased the population of bifidobacterium species [197]. In addition, several dietary intervention studies in healthy individuals or in obese subjects showed that polyphenols enhanced the population of SCFA-producing bacteria. In this regard, it was shown that oral intake of pomegranate polyphenols (1000 mg/day for 4 weeks), which is a rich source of tannins, anthocyanins, and non-colored flavonoids, as well as lignans [198], increased the population of *A. muciniphila* in the fecal samples of healthy subjects [199]. Daily intake of 272 mL red wine (equivalent to 798 mg phenols, including 104.68 mg flavan-3-ols, 26.44 mg anthocyanins, and 21.49 mg hydroxybenzoic acids) for 15 days increased the proportion of *F. prausnitzii* in feces of obese [200], but did not alter its profile in healthy subjects [201], suggesting that grape phenolic compounds may exerts beneficial effects in humans via modulating gut bacteria composition. Relative to limited studies in humans, more research on the effect of polyphenols on gut bacteria has been done in rodent models. It was shown that feeding mice HFD containing 0.1% oolong tea polyphenols increased the proportion of *F. prausnitzii* and other SCFA-producing bacteria, concomitant with increase in the fecal contents of SCFAs in mice [202]. Similarly, oral intake of mango (Mangifera indicaL. cv. “Keitt”) polyphenol extracts for 9 weeks increased the butyrate-producing bacteria, *Clostridium butyrium*, and improved butyrate production by 150% in rats [203]. However, some studies reported inhibitory or no effect of polyphenols on the gut bacteria. Daily intake of 10 g/kg black tea or green tea polyphenols-containing diets, which contain 9.3% and 65.9% catechins, respectively, for 3 weeks reduced the abundance of *Clostridium* XIVa (butyrate producer) in the feces of rats, but had no significant effect on the cecal SCFA contents [204]. Dietary provision of 0.3% EGCG, one of the key polyphenols of green tea extract, reduced the abundance of fecal *Clostridial* IV and XIVa bacteria in rats, but had no effect on cecal SCFAs [205]. Most of the polyphenol extracts from beverages and fruits in the above-mentioned studies differ in polyphenol contents, compositions of polyphenolic compounds, and purity, which make it difficult to compare the effect of these polyphenol extracts on microbiota or microbial products. Furthermore, fecal samples are often used for analysis of gut microbiota population, but both cecal and fecal samples have been used for the analysis of SCFA production. However, the population of microbiota from cecum and feces are different, in particular, the Clostridium group, which has a lower proportion in cecal microbiota than in fecal microbiota [206]. Therefore, it is possible that the difference in the microbiota population is responsible for the observed difference in SCFA contents between feces and cecum in the same subject. In this regard, comparison between studies using different sample origins might lead to inconsistent outcomes.

### 3.3. Polyphenols and DPP-IV

In addition to directly stimulating GLP-1 secretion, some polyphenols may suppress DPP-IV activity [207] (Table 3), thus potentially increasing the half-life of GLP-1 in the circulation. For example, it was shown that bis-pyrano prenyl isoflavone isolated from leaves of P. *molluginifolia* inhibited 53% DPP-IV activity at 100 µM in a cell-free assay [208]. In addition, rutin, eriocitrin, eriodictyol, naringenin, naringin, hesperidin, and hesperetin exhibited inhibitory effects on DPP-IV activity in the cell-free assay, with the IC50 of 0.485, 5.44, 3.91, 5.5, 3.82, 5.7, and 5.7 mM, respectively [209]. Interestingly, the potency of polyphenols on inhibiting DPP-IV could vary between the species from which the enzyme is derived. For example, apigenin exhibited potent inhibition of porcine kidney DPP-IV with an IC50 at only 0.14 ± 0.02 μM [207], but it is much less potent in inhibiting recombinant human DPP-IV, with a 200 μM concentration only ablating the enzyme activity by 44% [210]. Although the exact reason for this difference is not clear, it could be due to the difference in DPP-IV amino acid sequence between different mammalian species. While the amino acid sequences of DPP-IV are highly conserved, porcine DPP-IV only shares 88% identity in its amino acid sequence with that of human DPP-IV [211]. Given this possibility, the species of DPP-IV should be considered in determining the effects of polyphenolic compounds on its activity. Physiologically, there are two forms of DPP-IV, a soluble form in the blood and a membrane-anchored form on the luminal surface of the vascular endothelium. Both forms of DPP-IV have identical catalytic region, but the soluble form of DPP-IV lacks the cytoplasmic and transmembrane domains [212]. Therefore, polyphenolic compounds could have different interaction modes with two forms of DPP-IV that may result in different inhibitory efficacy. A study showed that a single dose of 1 g/kg of GSPE via oral gavage reduced intestinal DPP-IV activity (34.4%), whereas it did not inhibit plasma DPP-IV activity [213]. It is possible that the relatively low inhibitory effect on the circulating DPP-IV could be due to the low bioavailability of polyphenolic compounds. It may also be caused by the fact that the majority of polyphenols and their metabolites (over 80%) in the blood are already associated with proteins, primarily albumin [210], which may prevent their further interaction with DPP-IV. Interestingly, a study showed that GSPE effectively inhibited the activity of recombinant human DPP-IV in a cell-free system, achieving 70% inhibition at the concentration of 200 mg/L, but this dose of GSPE only exerted a weak inhibitory action on the activity of DPP-IV that was isolated from human plasma [213]. Therefore, DPP-IV proteins from the blood should be used to better resemble the physiological environment for assessing the inhibitory action of polyphenols in vivo.

In addition to the sources of DPP-IV, the substrates used in DPP-IV activity assay could confound interpretation of the results. Three different substrates are used in the DPP-IV assay, Gly-Pro-p-nitroanilide hydrochloride in colorimetric assay, Gly-Pro-7-amido-4-methylcoumarin hydrobromide (GP-AMC) in fluorometric assay, and Gly-Pro-aminoluciferin in luminescent assay. The activity of DPP-IV was 10-fold higher using the fluorometric substrate compared to the colorimetric substrate, and it is even higher using luminescent substrate as compared to the fluorometric substrate [214]. For example, it was shown that flavone apigenin at 200 μM only inhibited DPP-IV activity by 20% using a fluorometric assay [210], but its value of IC_50_ for inhibiting DPP-IV was 0.14 μM using the luminescent substrate [207]. While the reasons for the discrepancy between these results are unclear, there is possibility that polyphenols may interact with colorigenic, fluorogenic, and/or luminescent substrates to alter the assay sensitivity. Indeed, it was shown that some flavonoids (cyanidin, pelargonidin, catechin, myrecetin and kaempferol) bind to proline residue [215], which could modulate the availability of these compounds for interaction with DPP-IV. Interestingly, a more recent study showed that coumarin, a component of GP-AMC, inhibits DPP-IV activity [216]. However, it is unknown whether GP-AMC also has ability to inhibit DPP-IV. Therefore, in determining the inhibitory effect of polyphenolic compounds on DPP-IV, the type of assay methods, the sources and species of the enzymes, and inclusion of positive control should all be carefully considered.

It appears that DPP-IV can be inhibited by a great number of structurally diverse compounds, ranging from a simple phenolic compound, such as gallic acid, to polyphenolic flavonoid glycosides, such as eriocitrin [207]. DPP-IV has three major active sites (S1, S2, and S3) for its enzymatic activity, including the side chains of a catalytic triad (Ser630, Asn710, and His740) in the S1, cavity near Glu205, Glu206, and Tyr662 in S2, and Ser209, Arg358, and Phe357 in S3 [217]. The inhibitory action of polyphenolic compounds in DPP-IV might be due to the formation of hydrogen bonds between the amino acid residues in the catalytically active sites of DPP-IV and the hydroxyl groups of polyphenols. Indeed, in vitro assay showed that resveratrol (100 µM) has an inhibitory effect on DPP-IV activity [218] and molecular docking analysis demonstrated that resveratrol can form a hydrogen bond between its hydroxyl groups and the hydroxyl group of the side chain of DPP-IV at Ser630 (S1) and Ser209 (S3) [207]. In addition, resveratrol may also partially inhibit DPP-IV activity by forming electrostatic interactions with DPP-IV active sites Glu205 and Glu206 (S2) [207]. Based on computer-based molecular docking study, flavonoids are also potential DPP-IV inhibitors. The common characteristics of binding mode between flavonoids such as luteolin and apigenin and DPP-IV include the π-interaction between the benzene ring of flavonoids and the benzene amino acids at the active sites of DPP-IV and a hydrogen bond between the oxygen on the C ring of flavonoids and the ammonium of a guanidine group of Arg358 in the active site S3 of DPP-IV [207]. It was found that chrysin, a flavone glycoside, can inhibit DPP-IV due to the π-interaction between the benzene ring of chrysin and the S1 pocket (Tyr547) of the enzyme [219]. However, the results of molecular docking analyses from different research groups were not all consistent. Fan et al. showed that quercetin and genistein suppress DPP-IV activity by forming hydrogen bond [207], but Srivastava et al. found that quercetin and genistein have no effect on DPP-IV activity [220]. The inconsistency in the findings might be caused by using various crystal structures of DPP-IV. Indeed, Fan et al. and Srivastava et al. used different DPP-IV structures for molecular docking analysis. In addition, the diverse algorithm parameters in docking software and the lack of the validation of prediction performance could lead to the inconsistent results. Thus, validation of the results from molecular docking analysis using enzyme and cells are essential for confirming the efficacy of polyphenols in inhibiting DPP-IV.

## 4. Conclusions

Loss of functional β-cell mass plays a central role in the deterioration of blood glucose control in T2D. The GLP-1 signaling pathway has been extensively explored for developing incretin-based therapies for T2D, as its activation can promote GSIS and β-cell mass. Polyphenols are a group of plant-derived compounds that may exert beneficial effects on human health. Emerging evidence shows that some phenolic compounds can induce GLP-1 secretion from intestinal L-cells, and thus may be helpful in improving glucose homeostasis. However, this is still largely an unexplored territory and there remain many interesting questions that need to be addressed. It is unclear which category of phenolic compounds in the diet is superior to the others in stimulating GLP-1 secretion. Therefore, studies should be carried out to compare the efficacy of different categories of polyphenolic compounds, which may provide a better understanding of the chemical structure-function relationship, and thus shed light on how a particular polyphenolic compound activates GLP-1 secretion. Given that most of the non-absorbed polyphenols were fermented by gut microbiota and some products derived from fermentation was found to induce GLP-1 secretion, it is appealing to investigate whether and how the gut bacteria play a role in mediating polyphenol regulation of GLP-1 secretion.

## Figures and Tables

**Figure 1 molecules-26-00703-f001:**
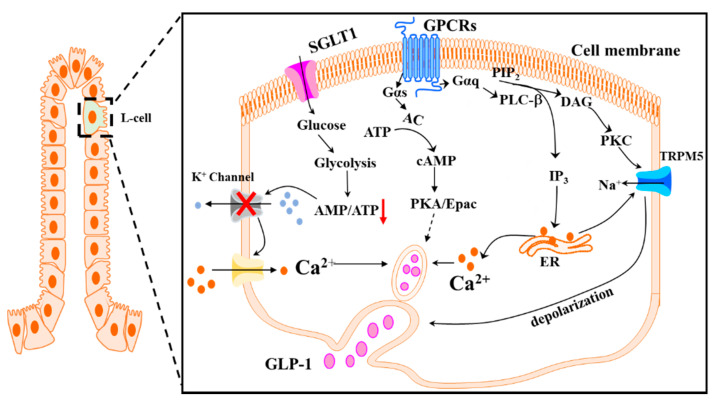
Schematic overview of nutrient-induced GLP-1 secretion from L-cells. AC: adenylyl cyclase; DAG: diacylglycerol; GLP-1: glucagon-like peptide-1; GPCRs: G-protein coupled receptors; IP3: inositol trisphosphate; PLC-β: phospholipase C-β; PKA: protein kinase A; PKC: protein kinase C; PIP2: phosphatidylinositol 4,5-biphosphate; SGLT1: sodium/glucose co-transporter 1; TRPM5: transient receptor potential channel M5.

**Figure 2 molecules-26-00703-f002:**
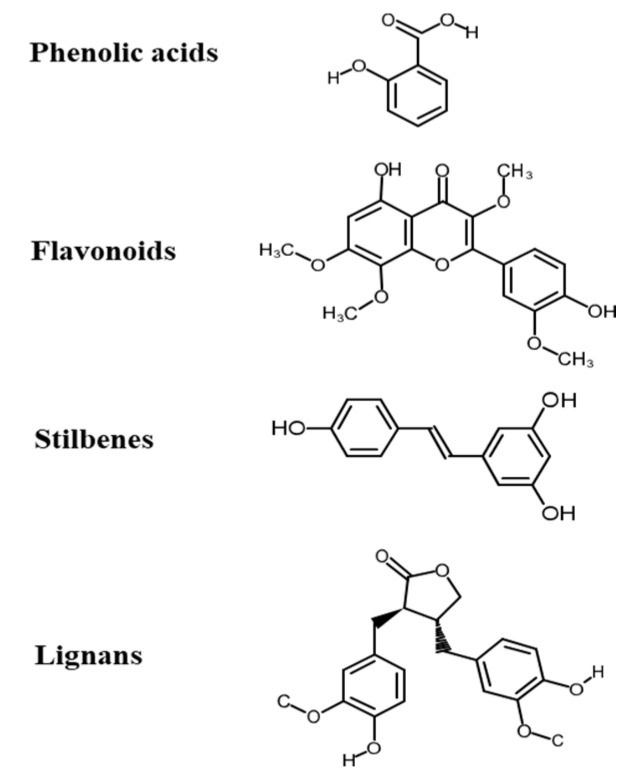
Chemical structures of polyphenols from four categories.

**Table 1 molecules-26-00703-t001:** Polyphenolic compounds that regulate GLP-1 secretion via G protein coupled receptor (GPCR)- or cAMP- mediated mechanism.

Polyphenolic Compounds	Chemical Structure	Dose	Model	Effect on GLP1	Possible Mechanisms
Curcumin [125,127]	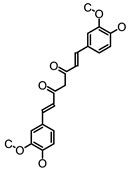	10 μM 1.5 mg/kg b.w.	Glutag L-cells, Rat	Increase GLP-1 secretion; increase plasma GLP-1 level	GPR40/120-dependent pathway
Delphinidin 3-rutinoside [128]	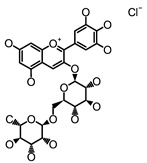	10 μM	Glutag L-cells	Increase GLP-1 secretion	GPR40/120-dependent pathway
EGCG [145,146,147]	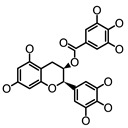	300 μM 500 mg	Mouse ileum segment, diabetic patients	Increase GLP-1 secretion; increase plasma GLP-1 level	Tas2Rs-dependent pathway
Genistein [165]	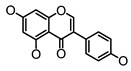	20 mg/kg b.w.	Alloxan-induced insulin deficient diabetic rats [165]	Increase tissue content of GLP-1	cAMP signaling by activating AC
Grifolic acid [132]	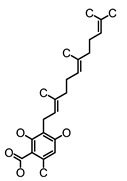	30 μM	STC-1 cells	Increase GLP-1 secretion	GPR120/Ca^2+^ signaling
Grifolic acid methyl ether [132]	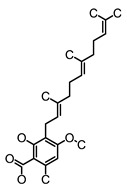	30 μM	STC-1 cells	Increase GLP-1 secretion	GPR120/Ca^2+^ signaling
Hispidulin [166]	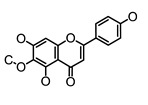	1 μM 20 mg/kg b.w.	Glutag L-cell, STZ-induced diabetic mice	Increase GLP-1 secretion; increase plasma GLP-1 level	Inhibiting PDE activity
Silymarin [169]	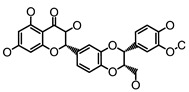	100 mg/kg b.w.	Obese rats	Increased serum GLP-1 level	Inhibiting PDE activity
Caffeoylquinic acid [173]	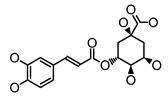	IC50: 0.49 mM	Human plasma platelets	Not applicable	Inhibiting PDE activity
Caffeic acid [173]	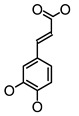	IC50: 0.48 mM	Human plasma platelets	Not applicable	Inhibiting PDE activity

**Table 2 molecules-26-00703-t002:** Polyphenols regulates short chain fatty acid (SCFA)-producing microbiota.

Polyphenols	Polyphenolic Composition	Dose	Model	Effects on Microbiota
Catechins [196]	(+)-Catechin	150 mg/mL	Ex vivo: human fecal microbiota	Increase the growth of *E. rectale*
Chlorogenic acid [197]	Chlorogenic acid	10 μg/mL	Ex vivo: human fecal microbiota	Increase the population of bifidobacterium species
Pomegranate polyphenols [199]	Not available	1000 mg/day	Healthy subjects	Increase the population of *A. muciniphila*
Red wine [200]	14% flavan-3-ols, 3.6% anthocyanins, 2.9% hydroxybenzoic acids	272 mL/day	Healthy and obese subjects	Increase the proportion of *F. prausnitzii* in obese subjects
Oolong tea polyphenols [202]	Catechins (88.79 μg/g)	0.1% oolong tea polyphenols in the high-fat diet	Mice	Increase the proportion of *F. prausnitzii* and other SCFA-producing bacteria, as well as increase in the fecal contents of SCFAs
Mango polyphenol extracts [40]	Gallic acid (7.35 mg/L)	ad libitum	Rats	Increase *Clostridium butyrium*, and improve butyrate production
Black tea or green tea polyphenols [204]	Catechines (275 g/kg in green tea, and 35 g/kg in black tea)	10 g/kg/day	Rats	Reduce the abundance of *Clostridium* XIVa
EGCG [205]	EGCG	0.3% in the diet	Rats	Reduce the abundance of fecal *Clostridial* IV and XIVa bacteria

**Table 3 molecules-26-00703-t003:** Polyphenols regulates dipeptidyl peptidase IV (DPP-IV) activity.

Polyphenolic Compounds	Effective Concentration	Detection Assay	Effect on DPP-IV Activity	Possible Mechanism
Bis-pyrano prenyl isoflavone [208]	100 μM	DPP-4 fluorometric assay	53% inhibition	Not available
Rutin [209]	IC50: 0.485 mM	DPP-4 fluorometric assay	50% inhibition	Not available
Eriocitrin [207,209]	IC50: 5.44 mM; 10.36 ± 0.09 μM	DPP-4 fluorometric assay	50% inhibition	H Bonds and π interactions
Eriodictyol [209]	IC50: 3.91 mM	DPP-4 fluorometric assay	50% inhibition	Not available
Naringenin [207,209]	IC50: 5.5 mM; 0.24 ± 0.03 μM	DPP-4 fluorometric assay	50% inhibition	H Bonds and π interactions
Naringin [209]	IC50: 3.82 mM	DPP-4 fluorometric assay	50% inhibition	Not available
Hesperidin [209]	IC50: 5.7 mM	DPP-4 fluorometric assay	50% inhibition	H Bonds and π interactions
Hesperetin [207,209]	IC50: 5.7 mM; 0.28 ± 0.07 μM	DPP-4 fluorometric assay	50% inhibition	H Bonds and π interactions
Apigenin [207,210]	IC50: 0.14 ± 0.02 μM; 200 μM	DPP-IV Glo^TM^ Protease Assay	50% inhibition	H Bonds and π interactions
Quercetin [207]	IC50: 2.92 ± 0.68 μM	DPP-IV Glo^TM^ Protease Assay	50% inhibition	H Bonds and π interactions
Genistein [207]	IC50: 0.48 ± 0.04 μM	DPP-IV Glo^TM^ Protease Assay	50% inhibition	H Bonds and π interactions
Resveratrol [207,218]	IC50: 0.0006 ± 0.0004 μM	DPP-IV Glo^TM^ Protease Assay	50% inhibition	H Bonds
Gallic acid [207]	IC50: 4.65 ± 0.99 μM	DPP-IV Glo^TM^ Protease Assay	50% inhibition	H Bonds and π interactions

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
