# Peer review of "The Emerging Role of Polyphenols in the Management of Type 2 Diabetes"

_molecules, 2021, doi:10.3390/molecules26030703_

Round 1

Reviewer 1 Report

The article, entitled The Emerging Role of Polyphenols in the management of type 2 diabetes , represents a good summary of what is reported in the literature on this subject. The positive antioxidant effects of polyphenols are well known, and are also associated to the type 2 diabetes and to the importance of microbiota. There are also many recent reviews on these topic. However the article even if lacks in part of novelty, is well written and reports many bibliographic references.

Author Response

We are grateful for the  reviewer's time reviewing this manuscript with positive comments. Even there are no suggested revisions and edits, we have carefully addressed each of the points raised by other reviewers . Further, we have thoroughly reviewed the manuscript and corrected grammatical and typographical errors. Finally, the manuscript is formatted according to the journal guideline. Overall, we think that the manuscript is further improved and hope that these revisions are satisfactory and the manuscript can be accepted for publication.

Reviewer 2 Report

Manuscript Number:         molecules-1019086

Title: The Emerging Role of Polyphenols in the management of type 2 diabetes

Comments to Authors

Recommendation:          Reject

The article “The Emerging Role of Polyphenols in the management of type 2 diabetes” gave an overview of nutritional regulation of GLP-1 secretion, pointing out the roles of polyphenols in GLP-1 secretion and degradation, discussing the effects of polyphenols on microbiota and microbial metabolites that could indirectly modulate GLP-1 secretion.

Although the interesting theme, thoroughly explanation of regulation process of the GLP-1 secretion in the first part of the paper, some aspects of the introduction and Part 3 need considerable improvement.

In the present form, the manuscript did not meet the Journal's request.

  1. In Introduction section, authors should provide better organisation of the data that are presented.
  2. When mentioning the source of polyphenols, although the aim is not to give the plants’ sources of the polyphenols, if authors decide to give the common name of the plant, it is obligatory to give full Latin name of the source, as well (the name of species, author name and family name, mandatory, page 2, lines 57, 76,…387, 388….etc), especially as authors partly named the polyphenols exhibiting the certain activity and partly giving the name of the plants rich in polyphenols, not stating clearly  the type of the polyphenols.
  3. The title gave the impression that authors would pay attention to specific compounds, but throughout the text sometimes it is difficult to identify the type of the polyphenols, as authors just gave the common information: ”the polyphenols from apple…coffee polyphenols…turmeric polyphenols ….”etc. what should be corrected by naming the polyphenols investigated in the cited literature.
  4. Please, give the literature (page 2, lines 65-69)
  5. Please, check throughout the text that abbreviations are correctly introduced
  6. The main concerns regarding the presented results refer to the authors’ presentation of the polyphenol compounds and its chemical characteristics. A lot of date is given, but without the neat organization, and overall impression is that there is a disorder in presenting data. The authors covered partly the information regarding the biosynthesis of the polyphenols present in the plants – they mentioned only shikimate/phenylpropanoid pathway as the biosynthesis pathway of “..some polyphenols…” , but further in the text they introduced the secoiridoides (oleuropein), from monoterpene group of the secondary metabolites, which did not comply with the mentioned biosynthesis Please, make sure that all mentioned polyphenols are classified into corresponding groups of the secondary metabolites, in regard to their biosynthesis pathways.
  7. Page 7, line 277-279 – the statement is contradictory
  8. Page 7, line2 84-285 – it is unclear if CQA derivatives is the only phenylpropanoids responsible for the mentioned activity
  9. The section 3.1, as introduction into different activity of polyphenols, should be reorganised, giving the brief insight which different mechanism of regulation of GLP-1 secretion would be discussed, instead of giving the examples of a few polyphenols. The table 1 should be expanded with each mentioned compound in the text, from subsection 3.1.1, 3.1.2
  10. In addition, the Table 1 could be expanded with the polyphenols mentioned in the section 3.2 and 3.3, or give another table with concise presentation of the specific compounds, experiments performed, mechanism of the action and the literature data used
  11. Please, explain what the numbers 1…10 in superscript Table 1 mean…
  12. Table 1 should contain the column with the references corresponding to the data given in the table regarding the polyphenols
  13. Please, give the data if the extracts mentioned page 8, lines 315 and 316 were the same (please give the main characteristic of the mentioned extracts – what does it mean: anthocyanin—rich extracts – were they characterized and standardized giving the percentage of the anthocyanin, and were the anthocyanins identified?
  14. Please, give the short description of the extracts used in the investigation – lines 315, 316?
  15. Page 8, lines 333-337: the given contemplation on different affinity of polyphenols for GPR120 based on only mentioned existence or lack of C2-C3 bond as the chemical differences between curcumin, as diarylheptanoid and flavonoids, as derivatives of benzo-γ-pyrone, could not be accepted – it does not possess the scientific soundness. If such a statement exists, please, provide the literature
  16. The authors should give the chemical structures of the all polyphenols mentioned in the text for which the mechanism of the action in regulation of GLP-1 secretion was explained.
  17. The name of the subsection 3.1.2 should be renamed
  18. Please, add the Family name of the plants’ species – page 9, line 388
  19. The text is full of insufficiently explained terms (what is the difference between Tas2R14, Tas2R5, Tas2R7, Tas2R39, and TasR243)? As being the receptors that might play significant role in activation of GLP-1 secretion, why this mechanism was not explained within section 2?
  20. What is the connection between FFAR1 and GPR40, ….etc and GPCRs….what does PIP2 mean? Please, the abbreviations should be introduced in the text as well, not only in the Legend of the Figure.
  21. Please, correct “..specie”….
  22. Please, specify the “…grape polyphenols…” (page 11, lines 458, 486), “…pomegranate polyphenols…” (page 11, line 483)
  23. Please, precise the “…black or green tea polyphenols containing diets…”, what kind of polyphenols
  24. Please, introduce the DPP-IV abbreviation
  25. In Section 2, the connection in GLP-1 and DPP-IV should be given
  26. There is no critical point of view hen authors represented the results of the cited authors’ investigation (page 12, line 546) – give the explanation WHAT might contribute to properly set the experimental conditions…

The overall impression is that authors, although presenting the extensive literature survey on up to date investigation on polyphenols effects on GLP-1, did not give critical point of their view. The manuscript lacks in organisation. The important facts are not explained correctly, a lot of shortcomings made the interesting theme confusing. Although aiming in plants' polyphenols role in the management of type 2 diabetes, none of chemical structure had not been presented.

Author Response

1. In Introduction section, authors should provide better organisation of the data that are presented.
Response: While We are not so sure about what was suggested specifically, we reviewed this section and made some changes in that regard.

2. When mentioning the source of polyphenols, although the aim is not to give the plants’ sources of the polyphenols, if authors decide to give the common name of the plant, it is obligatory to give full Latin name of the source, as well (the name of species, author name and family name, mandatory, page 2, lines 57, 76,…387, 388….etc), especially as authors partly named the polyphenols exhibiting the certain activity and partly giving the name of the plants rich in polyphenols, not stating clearly the type of the polyphenols.                                              Response: Thank the reviewer for this suggestion. We added the full Latin name of the plants that presented in this review.

3. The title gave the impression that authors would pay attention to specific compounds, but throughout the text sometimes it is difficult to identify the type of the polyphenols, as authors just gave the common information: ”the polyphenols from apple…coffee polyphenols…turmeric polyphenols ….”etc. what should be corrected by naming the polyphenols investigated in the cited literature.
Response: Thanks for your comments. as studies on the effects of polyphenolic compounds on GLP-1 secretion are still very limited, we also included available studies in which a mixture of polyphenols but no specific polyphenolic compound were tested. Therefore, it is impossible for us to identify and name the polyphenols responsible for the stimulatory effect on GLP-1. Nevertheless, we think that the review on this topic is timely, which provide emerging evidence that polyphenols can induce GLP-1 secretion in the gut.

4. Please, give the literature (page 2, lines 65-69)
Response: The reference has been provided.

5. Please, check throughout the text that abbreviations are correctly introduced
Response: Thank the reviewer for this suggestion. We have checked all abbreviations and added an abbreviation list at the end of the main text.

6. The main concerns regarding the presented results refer to the authors’ presentation of the polyphenol compounds and its chemical characteristics. A lot of date is given, but without the neat organization, and overall impression is that there is a disorder in presenting data. The authors covered partly the information regarding the biosynthesis of the polyphenols present in the plants – they mentioned only shikimate/phenylpropanoid pathway as the biosynthesis pathway of “..some polyphenols…” , but further in the text they introduced the secoiridoides (oleuropein), from monoterpene group of the secondary metabolites, which did not comply with the mentioned biosynthesis. Please, make sure that all mentioned polyphenols are classified into corresponding groups of the secondary metabolites, in regard to their biosynthesis pathways.
Response: We thank the reviewer for this constructive suggestion. We added biosynthesis pathways of other polyphenols in the text, which are classified into the respective groups accordingly. By the way, secoiridoid polyphenols, typically derived from Oleaceae family, are synthesized through a mevalonic acid pathway (reference 104).

7. Page 7, line 277-279 – the statement is contradictory
Response: We reorganized this section and revised this statement.

8. Page 7, line2 84-285 – it is unclear if CQA derivatives is the only phenylpropanoids responsible for the mentioned activity
Response: I agree with the reviewer that it is unknown whether CQA derivatives are the only phenylpropanoids responsible for the effect on GLP-1 secretion, and no further published studies as to our knowledge to examine the effects of the individual compounds on GLP-1 secretion.

9. The section 3.1, as introduction into different activity of polyphenols, should be reorganised, giving the brief insight which different mechanism of regulation of GLP-1 secretion would be discussed, instead of giving the examples of a few polyphenols. The table 1 should be expanded with each mentioned compound in the text, from subsection 3.1.1, 3.1.2
Response: We thank the reviewer for this thoughtful suggestion. We reorganized the introduction of section 3.1 as suggested. Also, table 1 has been updated with chemical structures of the polyphenolic compounds included.

10. In addition, the Table 1 could be expanded with the polyphenols mentioned in the section 3.2 and 3.3, or give another table with concise presentation of the specific compounds, experiments performed, mechanism of the action and the literature data used.
Response: As suggested, we expanded the Table 1 with chemical structures of the polyphenolic compounds in the section 3.1. In addition, we added the Table 2 for the polyphenols presented in section 3.2 and Table 3 for those in section 3.3.

11. Please, explain what the numbers 1…10 in superscript Table 1 mean…
Response: These are cited references and are not correctly numbered. This error is corrected.

12. Table 1 should contain the column with the references corresponding to the data given in the table regarding the polyphenols
Response: A column with the references is now added.

13. Please, give the data if the extracts mentioned page 8, lines 315 and 316 were the same (please give the main characteristic of the mentioned extracts – what does it mean: anthocyanin—rich extracts – were they characterized and standardized giving the percentage of the anthocyanin, and were the anthocyanins identified?
Response: The composition of anthocyanin is provided, which contains 19.3% D3R, 4.6% delphinnidin 3‐glucoside, 19.4% cyanidin 3‐rutinoside, and 2% cyanidin 3‐glucoside.

14. Please, give the short description of the extracts used in the investigation – lines 315, 316?
Response: We provided the polyphenolic composition, which contains 19.3% D3R, 4.6% delphinnidin 3‐glucoside, 19.4% cyanidin 3‐rutinoside, and 2% cyanidin 3‐glucoside.

15. Page 8, lines 333-337: the given contemplation on different affinity of polyphenols for GPR120 based on only mentioned existence or lack of C2-C3 bond as the chemical differences between curcumin, as diarylheptanoid and flavonoids, as derivatives of benzo-γ-pyrone, could not be accepted – it does not possess the scientific soundness. If such a statement exists, please, provide the literature
Response: The literature for this information is now provided (line 379).

16. The authors should give the chemical structures of the all polyphenols mentioned in the text for which the mechanism of the action in regulation of GLP-1 secretion was explained.
Response: The chemical structures were added in the table 1.

17. The name of the subsection 3.1.2 should be renamed
Response: The title of subsection 3.1.2 has been renamed as “Polyphenols as regulators of cAMP signaling.”

18. Please, add the Family name of the plants’ species – page 9, line 388
Response: The family name of the plants has been provided, “(Fabaceae family)”.

19. The text is full of insufficiently explained terms (what is the difference between Tas2R14, Tas2R5, Tas2R7, Tas2R39, and TasR243)? As being the receptors that might play significant role in activation of GLP-1 secretion, why this mechanism was not explained within section 2?
Response: Bitter taste receptors (Tas2Rs) consist of 25 members in humans and 35 members in rats and mice, all of which are GPCRs. The TAS2Rs are presented in taste cells and gastrointestinal tracts to detect a diversity of bitter compounds and transduce the signal through G-protein-coupled signaling. We added more information to define some of these receptors as relevant and the underlying mechanism for mediating GLP-1 secretion (Lines 285-388, and line 313, 314, 324).

20. What is the connection between FFAR1 and GPR40, ….etc and GPCRs….what does PIP2 mean? Please, the abbreviations should be introduced in the text as well, not only in the Legend of the Figure.
Response: FFA1 is also known as GPR40, and we clarified this and others where applicable. We now have all abbreviations defined throughout the text.

21. Please, correct “..specie”….
Response: This typo is now corrected.

22. Please, specify the “…grape polyphenols…” (page 11, lines 458, 486), “…pomegranate polyphenols…” (page 11, line 483)
Response: Grape polyphenols primarily include primarily include anthocyanins, flavanols, flavonols, stilbenes and phenolic acids (Reference 195, line 540). In addition, major polyphenols contained in red wine used in cited study were also included (lines 566-568). While the specific constituents of pomegranate polyphenols used in the cited article are not provided, we provided information about the composition of major pomegranate polyphenols (lines 565-566).

23. Please, precise the “…black or green tea polyphenols containing diets…”, what kind of polyphenols
Response: We provided available information presented in this study.
24. Please, introduce the DPP-IV abbreviation
Response: The abbreviation of DPP-IV was provided in page 2 ,line 88.

25. In Section 2, the connection in GLP-1 and DPP-IV should be given
Response: GLP-1 has a very short half-time (2-3 minutes) in the blood because it is rapidly degraded by DPP-IV that cleaves GLP-1 from its N-terminal into GLP-1-(9-37) and GLP-1-(9-36) NH2 [43]. We have this information added to the Section 2 of the revised manuscript.

26. There is no critical point of view hen authors represented the results of the cited authors’ investigation (page 12, line 546) – give the explanation WHAT might contribute to properly set the experimental conditions…
Response: I wish I could figure out what the reviewer asked for here. Please explain further and thanks.

Reviewer 3 Report

The Emerging Role of Polyphenols in the 3 management of type 2 diabetes is a review article that addresses the role of phenolic compounds in diabetes control.

Among the various possible mechanisms, it focuses on the control of the glucagon-like hormone peptide-1 (GLP-1) which is very important in glucose homeostasis.

Thus, it begins by explaining how the regulation of GLP-1 secretion is made at the intestinal level, by food (glucose, fatty acids and high doses of proteins) but also by neuronal and endocrine regulators.

Then the authors describe how polyphenols can stimulate the secretion of GLP-1, namely through the intervention of the intestinal microbiota. Not only because polyphenols can modulate the abundance of the gut microbiota, but also, because many of these compounds are not absorbed and are fermented by various colonic microbiota to generate a wide range of metabolites that have physiological actions.

The manuscript reviews several studies with polyphenols in foods that showed an improved plasma GLP-1 concentration, accompanied with reduced hyperglycemia, as well as the doses that showed the effect, and how polyphenols could modulate enzymes and hormones, particularly in gastrointestinal tract.

The authors make a critical analysis that the doses used in cell tests may not be effective in humans due to the low bioavailability of these compounds.

In my opinion the review is interesting, well organized and with many examples of results published in a relatively recent bibliography. The topic is also important because on the increasing importance of gut microbiota in some diseases, and how action on microbiota can be a strategy to help the resolution of some diseases

Author Response

(The authors gave the same response as above.)

Round 2

Reviewer 2 Report

Comments on revised manuscript

Manuscript No: molecules-1019086

Recommendation: Minor revision

The paper in the revised form is acceptable for publishing.

As just one point remained to be addressed, due to unclearness of the request (point 26, I wish I could figure out what the reviewer asked for here. Please explain further and thanks, in revised manuscript page 15, lines 557-563 ), the authors are asked to give their or from literature cited explanation why polyphenolic compounds exhibited different inhibitory effects on DPP-IV activity when different substrates were used (given example in the paper was fluorometric compared to colorimetric substrate). What were the requirements that should be fulfilled to choose adequate type of assay methods and sources and species of the enzymes to be able to accurately determine the investigated activity of certain polyphenolic compounds? 

Author Response

Thank you very much for taking your valuable time to review our manuscript again, and also for raising this very interesting and critical question.

We tried to address this issue as to the best of our knowledge and please see the lines 560-568.